# Lower Critical Solution Temperature Tuning and Swelling Behaviours of NVCL-Based Hydrogels for Potential 4D Printing Applications

**DOI:** 10.3390/polym14153155

**Published:** 2022-08-02

**Authors:** Shuo Zhuo, Elaine Halligan, Billy Shu Hieng Tie, Colette Breheny, Luke M. Geever

**Affiliations:** 1Material Research Institute, Technological University of the Shannon, Midlands Midwest, Dublin Road, N37 HD68 Athlone, County Westmeath, Ireland; a00218425@student.ait.ie (E.H.); a00248520@student.ait.ie (B.S.H.T.); cbreheny@ait.ie (C.B.); 2Applied Polymer Technologies Gateway, Material Research Institute, Technological University of the Shannon, Midlands Midwest, Dublin Road, N37 HD68 Athlone, County Westmeath, Ireland

**Keywords:** NVCL, photopolymerisation, LCST tuning, pulsatile swelling studies

## Abstract

The phase transitions of poly (N-vinyl caprolactam) (PNVCL) hydrogels are currently under investigation as possible materials for biomedical applications thanks to their thermosensitive properties. This study aims to use the photopolymerisation process to simulate the 4D printing process. NVCL-based polymers with different thermal properties and swellability were prepared to explore the possibility of synthetic hydrogels being used for 4D printing. In this contribution, the thermal behaviours of novel photopolymerised NVCL-based hydrogels were analysed. The lower critical solution temperature (LCST) of the physically crosslinked gels was detected using differential scanning calorimetry (DSC), ultraviolet (UV) spectroscopy, and cloud point measurement. The chemical structure of the xerogels was characterised by means of Fourier transform infrared spectroscopy (FTIR). Pulsatile swelling studies indicated that the hydrogels had thermo-reversible properties. As a result, the effect of varying the macromolecular monomer concentration was apparent. The phase transition temperature is increased when different concentrations of hydrophilic monomers are incorporated. The transition temperature of the hydrogels may allow for excellent flexibility in tailoring transition for specific applications, while the swelling and deswelling behaviour of the gels is strongly temperature- and monomer feed ratio-dependent.

## 1. Introduction

Hydrogels are known as cross-linked hydrophilic polymer networks formed by the reaction of one or more monomers [1]. They have the capability of imbibing significant amounts of fluids from the aqueous media in which they are placed, causing hydrogels to swell, while retaining the overall structure [2]. Therefore, this kind of polymer material has novel development in various areas, including agriculture, sealing, coal dewatering, artificial snow, food additives, contact lenses, and biomedical products [3]. Environmentally sensitive hydrogels have also been referred to as ‘intelligent’ or ‘smart’ hydrogels [4] and have received increased interest in recent years. Smart hydrogels are known as polymers that possess the capability to abruptly change volume or properties in response to small external changes in environmental conditions [5]. This is owing to the characteristics of environmentally responsive polymers changing with the surrounding environment to meet the desired or specific requirements of different environmental conditions.

A well-studied example of smart materials is poly (N-vinyl caprolactam) (PNVCL), which is a temperature-responsive polymer. PNVCL has been utilised for synthesising hydrogels and has been used extensively in the development of smart drug delivery systems [6]. Because of the degradable performances, PNVCL has been incorporated with other additives or materials to form a new thermosensitive copolymer, with monomers such as poly(ethylene glycol) (PEG), vinyl acetate (VAc) [7], methacrylic acid (MAA) [8], and α-azide terminated poly(oligo(ethylene oxide) methyl ether methacrylate (N_3_-POEOMA) [9]. This series of copolymers were utilised in biomedical applications. NVCL polymer exhibits a phase transition and becomes less water-soluble as the temperature increases. This critical temperature is called lower critical solution temperature (LCST), which can be considered the critical point for dimensional expansion and contraction of the hydrogel samples. This feature provides the possibility of applying NVCL-based polymers to 4D printing development.

Photopolymerisation is a method to rapidly transform liquid monomers or macromere solution into crosslinked reliable networks. Inherent to this technology, a photoinitiator is necessary, which is a compound that undergoes a photoreaction on the absorption of radiation. This reaction is capable of initiating the polymerisation that results in significant changes in the physical properties of the suitable formulation [10]. Owing to the benefits of fast curing, high efficiency, solvent-free formulations, and low energy consumption, photopolymerisation is employed in a broad range of medical and biological applications, such as printing, dentistry, optical materials, encapsulating pancreatic islet cells, and blood vessel adhesives [11].

In simple terms, 4D printing can be achieved using a 3D printer to print environmentally sensitive polymers, which is a process of creating a physical object using appropriate additive manufacturing technology. It involves the construction of objects by laying down successive layers of stimuli-responsive composite or multi-material with varying properties. After being built, the object reacts to stimuli from the natural environment or through human intervention, resulting in a physical or chemical change in state over time [12]. Photopolymerisation has the same mechanism as the stereolithography (SLA) printing technique. Both of them use ultraviolet (UV) light to cure the objects. Therefore, in this study, smart NVCL-based hydrogels were synthesised using photopolymerisation to simulate the 4D printing program.

The most common materials used for achieving 4D printing are shape memory polymers (SMPs) thanks to their plasticity and simple operability [13,14,15,16]. However, little to no research has been conducted on the development of 4D parts using hydrogel-based NVCL copolymers. Therefore, the overall objective of the study was to examine the thermal and swelling properties of the prepared NVCL-based polymers so that the best NVCL hydrogel formulation could potentially be used for preparing smart 4D objects through a practical SLA 3D printing machine. Irgacure^®^ 2959 (Ciba Corp, New York, NY, United States) was chosen as a photoinitiator and used to initiate the reaction. N-vinylpyrrolidone (NVP) is a synthetic, non-ionic polymer that is soluble in water and in the form of an organic liquid. It is a highly amorphous polymer with a glass transition temperature of around 150 °C [17]. N,N-dimethyl acrylamide (DMAAm) is composed of two methyl groups that increase the electron density on the nitrogen atom and form the polarity on the amide structure, which causes the property of electron donation [18,19,20]. In this study, NVP and DMAAm monomers were incorporated because of their hydrophilic nature and high transition temperature [21]. Poly (ethylene glycol) dimethacrylate (PEGDMA) was chosen as a crosslinker in this work, which has attracted broad interest as a material for tissue engineering thanks to its biocompatibility, non-toxicity, and variable degradability, and it can be formed readily in situ [22]. The physically crosslinked polymers developed in this study possess various transition temperatures (LCSTs), and a number of chemically crosslinked hydrogels exhibit outstanding reversible swelling behaviours, which provide the possibility to achieve the 4D printing concept.

## 2. Materials and Methods

### 2.1. Materials

N-vinyl caprolactam (NVCL) was obtained from Sigma Aldrich Ireland with a molecular weight of 139.19 g/mol and a storage temperature from 2 to 8 °C. 4-(2hydroxyethoxy) phenyl-(2-hydroxy-2-propyl) ketone (Igracure^®^ 2959 Ciba Corp, New York, NY, United states) was obtained from Ciba Specialty Chemicals. N-vinylpyrrolidone (NVP) and N,N-dimethyl acrylamide (DMAAm) were purchased from Sigma Aldrich with a molecular weight of 111.14 g/mol and 99.13 g/mol, respectively. The chemical crosslinker used was poly (ethylene glycol) dimethacrylate (PEGDMA) supplied by Sigma Aldrich Ireland with a molecular weight of 550 g/mol. The chemical structures of these materials are listed in Table 1.

### 2.2. Hydrogel Synthesis

The hydrogels investigated in this study were prepared by free-radical polymerisation using UV light. These hydrogels were synthesised via physical and chemical crosslinking using a UV curing system (Dr. Gröbel UV- Elektronik GmbH, Ettlingen, Germany). This particular irradiation chamber is a controlled radiation source with 20 UV-tubes that provide a spectral range between 315 and 400 nm at an average intensity of 10–13.5 mW/cm^2^. The prepolymerised mixtures were prepared by combining the desired amounts of the monomer NVCL with specified amounts of other materials and 0.1 wt% photoinitiator and 2 wt% PEGDMA for chemical crosslinking gels. The compositions of the hydrogels are listed in Table 2. The batches were placed in a 100 mL beaker and mixed using a magnetic stirrer for 20 min until a homogeneous mixture was obtained. The solutions were pipetted into silicone moulds that contained disc impressions. Photopolymerisation was carried out for 10 min on each side, and all samples prior were dried to use for 24 h in a vacuum oven at 50 °C. In order to economise on material that possesses the lowest cytotoxic potential, the minimum amount of photoinitiator (0.1 wt%) was applied. The unpresented works performed in our lab found that homopolymer of NVCL with a 2 wt% crosslinker exhibited promising water absorption capabilities. Therefore, in this study, 2 wt% PEGDMA was included for the preparation of chemically crosslinked copolymers and terpolymers.

#### Preparation of Aqueous Solutions

Solutions of physically crosslinked samples were prepared by weighing appropriate amounts of the xerogel and distilled water (pH = 7.1), leaving these mixtures at room temperature until completely dissolved. These aqueous solutions were produced for subsequent use in LCST measurements. Based on the literature [7,23], the aqueous solution was prepared in a small tube with 1 g xerogel dissolved in 9 g distilled water to guarantee a 10 wt% concentration for all samples.

### 2.3. Attenuated Total Reflectance Fourier Transform Infrared Spectroscopy

Attenuated total reflectance Fourier transform infrared spectroscopy (ATR-FTIR) was carried out on a Perkin Elmer Spectrum One FT-IR Spectrometer (C-001), fitted with a universal ATR sampling accessory. All data were recorded at room temperature (<20 °C), in the spectral range of 4000–650 cm^−1^, utilising a 4-scan per sample cycle and a fixed universal compression force of 75 N. Subsequent analysis was performed using Spectrum software.

### 2.4. Differential Scanning Calorimetry

Differential scanning calorimetry studies were performed on all aqueous polymer solutions (TA instrument 2920 Modulated DSC, New Castle, DE, United States). The DSC was calibrated with indium standards. The physical crosslinked solution was pipetted into the aluminum pans and weighed out, ranging from 8 to 12 mg, using a Sartorius scale with 0.01 mg resolution. The empty sealed aluminum pans were used as the reference sample. All samples were examined under a pure nitrogen atmosphere at 30 mL/min. All samples were ramped from 10 °C to 60 °C at a rate of 1 °C/min. The results were plotted as a function of heat flow (W/g) against temperature (°C).

### 2.5. UV-Spectroscopy

The LCST behaviours of physically crosslinked samples were also investigated using a UV-spectroscopy synergy HT BioTek plate reader. The transmittance of samples in aqueous polymer solutions was measured at 500 nm at intervals of 1 °C in the range of 21–50 °C. The solution was allowed to equilibrate at each temperature for 20 min before the transmittance was measured.

### 2.6. Cloud Point Measurements

The LCST of polymer solutions can be determined using a visual cloud point method. A sealed glass test tube with a 75 mm path length was filled with selected polymer solutions and placed in a thermo-stable bath. The temperature was increased manually at a rate of 1 °C per 2 min, after the temperature reached a few degrees below the pre-estimated cloud point temperature. The LCST was recorded as the temperature at which the sample began to show initial signs of becoming turbid.

### 2.7. Pulsatile Swelling Studies

After photopolymerisation, the chemically crosslinked samples were placed in a vacuum oven for 24 h at 50 °C and the apparent dry weights (W_d_) were measured. The samples were tested at room temperature and 50 °C, below and above the LCST. Samples were placed in 30 mL of distilled water (pH = 7.1) to determine the swelling ratio. The percentage of gel swelling was calculated using Equation (1). All reported swelling studies were conducted in distilled water.
Swelling Ratio (%) = (W_t_ − W_d_)/W_d_ × 100%,(1)
where W_t_ and W_d_ are the weights of the gels in the swelling state at a predetermined time and the initial dry mass of the dried state, respectively.

## 3. Results and Discussion

### 3.1. Preparation of Samples

Photopolymerisation is not only a time- and material-saving processing method [11], but it also possesses the same mechanism as the SLA 3D printing technique. Therefore, in this work, physical and chemical hydrogels based on NVCL were synthesised via photopolymerisation to simulate a 3D printing process. As demonstated in Figure 1, each NVCL monomer consists of a tertiary amine with a carbonyl group and a methylene group, which has a double bond that separates during the photopolymerisation process in forming polymer chains with other monomers. The homopolymers, copolymers, and terpolymers NVP/NVCL, DMAAm/NVCL, and NVP/DMAAm/NVCL, respectively, were photopolymerised in the presence of 0.1 wt% Irgacure^®^ 2959 photoinitiator. PEGDMA is commonly used as a crosslinker, which serves to enhance the internal molecule network strength and was added into the solution to form the chemical crosslinked gels. Photoinitiators can decompose and generate free-radicals when exposed to a UV source, thereby initiating the polymerisation of the monomers. The disc samples were prepared based on Table 2. All samples were cured on a silicone moulding and dried for at least 24 h in a vacuum oven before use. As a result, all samples cured well and maintained their shape integrity. The xerogels were transparent and glass-like (Figure 2). This phenomenon indicated that the chains are arranged randomly and present no actual boundaries or discontinuities from which light can be reflected [24], thus indicating that the samples are amorphous and were successfully prepared.

### 3.2. Attenuated Total Reflectance Fourier Transform Infrared Spectroscopy

ATR-FTIR was used to determine if photopolymerisation was successfully carried out in NVCL-based polymers. Both physically and chemically crosslinked hydrogels were investigated using this technique. Table 3 summarises the band value of each chemical bond that exists in the monomer and synthetic samples. In NVCL-based copolymers, the monomers displayed characteristic absorption bands at 1625/1696 cm^−1^ (NVP), 1622/1662 cm^−1^ (NVCL), and 1608/1648 cm^−1^ (DMAAm). These bands are assigned to the carbonyl group C=O bond stretching vibration and the C=C bond stretching vibration. Furthermore, the C-C for the aliphatic group in the caprolactam ring was observed at 1428 cm^−1^, in accordance with the literature [25,26]. However, in the photopolymerised homopolymer and copolymer spectrum, only an intense peak at around 1620 cm^−1^ can be observed, attributed to the characteristic absorption of C=O stretching. The disappearance of the C=C bond peaks in the spectra indicates successful polymerisation of the copolymer [19,20], which was displayed by a red arrow in Figure 3. The absorbance bands at 2929 cm^−1^/2858 cm^−1^ for NVCL monomer, 2975 cm^−1^/2886 cm^−1^ for NVP, and 2937 cm^−1^ for DMAAm are attributed to the aliphatic C-H bending. Moreover, we observed the existence of bands at 2923 cm^−1^/2856 cm^−1^, 2926 cm^−1^/2859 cm^−1^, 2926 cm^−1^/2858 cm^−1^, 2926 cm^−1^/2859 cm^−1^, 2925 cm^−1^/2861 cm^−1^, 2928 cm^−1^/2864 cm^−1^, and 2926 cm^−1^/2862 cm^−1^ in C1, S1, S2, S3, S4, S5, and S6 samples, respectively. These findings correspond with another study [27]. While the those bands are discernible in monomers at 3105 cm^−1^/975 cm^−1^ (NVCL), 969 cm^−1^ (DMAAm) and 981 cm^−1^/843 cm^−1^ (NVP) have been previously reported to be associated with =CH or =CH_2_ bending [28]. The bands at 1480 cm^−1^ (NVCL), 1490 cm^−1^ (DMAAm), 1488 cm^−1^ (NVP), 1473 cm^−1^ (C1), 1471 cm^−1^ (S1), 1474 cm^−1^ (S2), 1473 cm^−1^ (S3), 1471 cm^−1^ (S4), 1483 cm^−1^ (S5) and 1473 cm^−1^ (S6) are attributed to the stretching of the C-N bond. The O-H bending that arises at 3449 cm^−1^, 3443 cm^−1^, 3453 cm^−1^, 3446 cm^−1^, 3420 cm^−1^, 3441 cm^−1^, and 3439 cm^−1^ in C1, S1, S2, S3, S4, S5, and S6, respectively, are the other pieces of evidence indicating that polymerisation was successful.

### 3.3. Phase Transition Determination for Physically Crosslinked Hydrogels

The LCST of temperature-responsive polymers plays a crucial role in designing smart 4D objects. The most important factors affecting the phase transition of PNVCL are copolymer composition and concentrations in aqueous media. In this section, the phase transition of physically crosslinked NVCL-based polymers was measured by employing three techniques to determine the LCST. The homopolymer samples showed a reversible phase transition in water at around 32 °C, which was in agreement with the literature [23]. By utilising each of these techniques, a common trend was established. Table 4 summarises the LCST values of pure physically crosslinked NVCL solutions and their copolymers using three techniques. In general, by alternating the feed ratio and adding the hydrophilic NVP and DMAAm monomers, NVCL-based copolymers or terpolymers were synthesised to have their own distinctive phase transition temperature. As the polymer chain contains increased hydrophilic constituent, the LCST becomes higher. The regularity obtained in this study was consistent with previous observations [29,30].

#### 3.3.1. Differential Scanning Calorimetry

DSC is a prevalent method for detecting the transition temperatures of samples. As the temperature rises slowly, it facilitates the separation of adjacent peaks or adjacent weightless platforms. In all cases, the onset value of the endothermal peak was used to establish the LCST value. The relatively strong hydrogen bonds formed between water molecules and N-H and C=O groups in dilute solutions became weaker and broke as the temperature increased, resulting in the endothermic heat of phase separation [29]. Illustrated in Figure 4 is a representative thermogram for control sample C1 (100PNVCL) homopolymer solution, where the onset value of LCST is shown as 32.3 °C. However, the LCST for copolymer samples S1 (30NVP/70NVCL), S2 (30DMAAm/70NVCL), and S3 (15NVP/15DMMAm/70NVCL) were 37.9 °C, 32.3 °C, and 36.3 °C, respectively. The thermograms of S1, S2, and S3 samples are illustrated in the Appendix A. As a result, the incorporation of DMAAm and NVP during polymerisation resulted in an increase in the LCST, which may be due to the stronger interaction between the hydrophilic segments of the polymer chains.

#### 3.3.2. UV Spectroscopy

The aqueous solution of NVCL-based copolymers and terpolymers was also investigated for LCST using UV-spectroscopy. This technique measures the absorbance of the samples. All samples were analysed over a temperature range of 22–50 °C. The image confined within Figure 5 illustrates the phase transition behaviour of NVCL-based polymers. The absorbance change point represents the LCST. The data change of the absorbance can be attributed to the transfer of the solution state, optically transparent below the LCST and opaque above the LCST. The inflection point in the S-shaped curve corresponds to the LCST. It can be seen in Figure 5 that the 10 wt% pure PNVCL solutions show a big increase when the temperature rises from 35 degrees to 36 degrees, which means that the phase transition has occurred. Additionally, it was noticed that the absorption rate S1 solution progresses towards zero in the absorbance spectrum, which means these samples were unable to experience the phenomenon from colourless to opaque within this limited range. As such, sample S1, containing 30% NVP and 70% NVCL, has an LCST of over 50 °C; sample S2, containing 30% DMAAm and 70% NVCL, has an LCST of 32 °C; whereas sample S3, which contains 15% NVP, 15% DMAAm, and 70% NVCL, has an LCST of 36 °C.

In the case of copolymers or terpolymers, a temperature increase is required for the phase change to be completed. This regularity can be explained by the fact that a small portion of the gel initially undergoes phase change at the onset temperature, while the majority gradually undergoes phase change as the temperature rises [31]. Owing to the different polymerisation kinetic constants of the respective monomers and the concentration of the mixture, the water content increases the hydrogen bonding reaction between the water and the polymer, which requires more thermal energy to break the water structure, resulting in a delayed phase transition [32].

#### 3.3.3. Cloud Point Measurements

The cloud point was defined as the temperature at which the first signs of turbidity appear in the solution. At an ambient temperature, the aqueous polymer solution was transparent, owing to the effect of the hydrophobic chains. The polymers were precipitating gradually, resulting in the solution becoming opaque upon the temperature increasing. At a specific temperature, the water becomes a poor solvent for the polymer, possibly owing to the new and less polar polymer conformation. Additionally, the hydrogen bonds were broken down by the growing internal energy, resulting in increased entropy in the system. In order to maintain the balance of entropy, the prevalence of hydrophobic molecules tended to aggregate, leading to phase separation [33]. It can be seen in Figure 6 that the samples show a reversible phase transition in water from transparent colourless solution to opaque as the temperature increased, which is similar to the results to the literature [34]. It was observed that the homopolymer C1 began to whiten at 32 °C, which is in accordance with the value achieved by the previous authors [35]. The cloud point analysis of copolymer samples resulted in an increase in LCST with the incorporating NVP and DMAAm monomers during photopolymerisation. The cloud points of S1, S2, and S3 samples were recorded at 49 °C, 33 °C, and 37 °C, respectively. This is consistent with the conclusions of the two previous experiments.

On the whole, the three techniques carried out in this study indicated that the LCST of the physically crosslinked samples can be increased by the combination of hydrophilic monomers, which offers flexibility and selectivity for a 4D-printed object when requiring use at different temperatures, such as for toys and temperature sensors.

### 3.4. Pulsatile Swelling Studies of Chemically Crosslinked Hydrogels

Owing to the presence of covalent bonds, the chemically crosslinked hydrogels are able to absorb water and swell. The 4D-printed objects require the ability to change shape or properties when a stimulus acts on them. Therefore, the swelling ability of such samples is the most critical factor in achieving 4D printing. A hydrogel-based sample placed in distilled water will absorb the solvent until the elasticity of the network and osmotic pressure are balanced. Swelling means water absorption is a characteristic property of hydrophilic groups in a polymer. The constitution and degree of crosslinking affect the swelling properties of the hydrogels, and the hydrogels’ mechanical properties decrease as the percentage of swelling increases [36]. Pulsatile swelling experiments were conducted to investigate the reversible response and thermo-sensitivity of these hydrogels to changes in environmental temperature. Pulsatile swelling studies were carried out on all chemically crosslinked circular discs above and below the LCST, which was determined by previous phase transition experiments. At various time intervals, samples were removed from distilled water and blotted with filter paper to absorb excess water on the surface, and then the samples were weighed and submerged into fresh distilled water until equilibrium was established. After that, the swelled samples were submerged into distilled water that was preheated to the required temperature and weighed in the same step before until a new equilibrium was achieved. When the temperature is below LCST, the hydrogen-bonding of the hydrophilic segments dominates the interaction between water and the polymer chains, and gel absorbs water and dissolves the polymer chains to form the swollen state. When the temperature increases above the LCST, the polymer chains shrink and expel water molecules through hydrophobic segments, which results in the precipitation of polymer and a globular state appears. Graphical results are exhibited in Figure 7, which displays the polymer swelling behaviour switch below and above LCST. As can be seen in Figure 7, Sample S4 (30NVP/70NVCL) can swell to seven times the weight of the original xerogel. However, the samples became soft and sticky after swelling, and it was easy to destroy their structure during the experiment. Whereas S5 (30DMAAm/70NVCL) and S6 (15NVP/15DMAAm/70NVCL) only expended five times the dry weight, the samples after absorbing water can also retain their toughness, shape, and integrity. For all S5 samples below LCST, the maximum swollen weight was reached after 24 h. For S4 and S6 copolymers, the maximum swollen weight was reached after 72 h. Subsequent to the maximum swollen weight being reached, when the temperature was above LCST, the samples s began to shrink and exhibit reversible shape change behaviour as each temperature rose and fell. As a result, for 30DMAAm/70NVCL samples, the swelling reaction time is at a minimum. In addition, the samples can swell to a similar size when the temperature drops again. The appearances of the S5 discs can be seen in Figure 8. In summary, the swelling properties were obviously affected by incorporating hydrophilic monomers. The S5 sample’s outstanding reversible swelling properties provide the possibility to change the final shape when applying 4D printing techniques.

## 4. Conclusions and Future Perspective

In summary, NVCL-based copolymers were successfully prepared using the photopolymerisation technique. This was confirmed with data via FTIR. The LCSTs of physically crosslinked hydrogels were determined by employing the differential scanning calorimetry, UV-spectroscopy, and cloud point measurement techniques. This study allows a much greater understanding of the LCST phase transition of novel NVCL-based copolymer systems, whose thermal properties were affected by the monomeric feed ratio. NVCL-based copolymers exhibited an increase in phase transition temperature depending upon the monomeric feed ratio. The swelling property of chemically crosslinked hydrogels was investigated by pulsatile swelling studies. The result showed that the swelling and deswelling behaviour of the gels is strongly temperature- and monomer feed ratio-dependent. Based on the findings of this study, the outstanding swellability and tunable phase transition temperature of the smart polymers were prepared, and the chemically crosslinked hydrogels exhibited the potential to be used in practical 4D printing techniques.

Further research will examine the feasibility of applying the formulations to a physical SLA 3D printer to achieve the actual 4D printing concept. Furthermore, in order to achieve more complex shapeshifting mechanisms, the incorporation of a number of different hydrogel materials is to be examined.

## Figures and Tables

**Figure 1 polymers-14-03155-f001:**
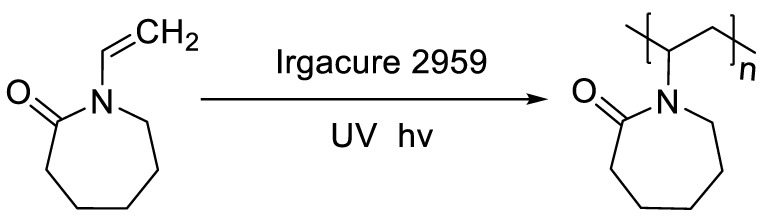
Scheme of photopolymerisation of N-vinylcaprolactam using Irgacure 2959 to yield poly (N-vinylcaprolactam).

**Figure 2 polymers-14-03155-f002:**
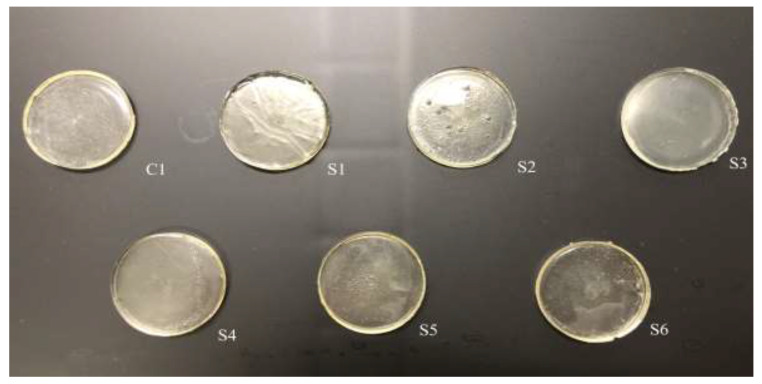
The appearance of the prepared disc xerogels.

**Figure 3 polymers-14-03155-f003:**
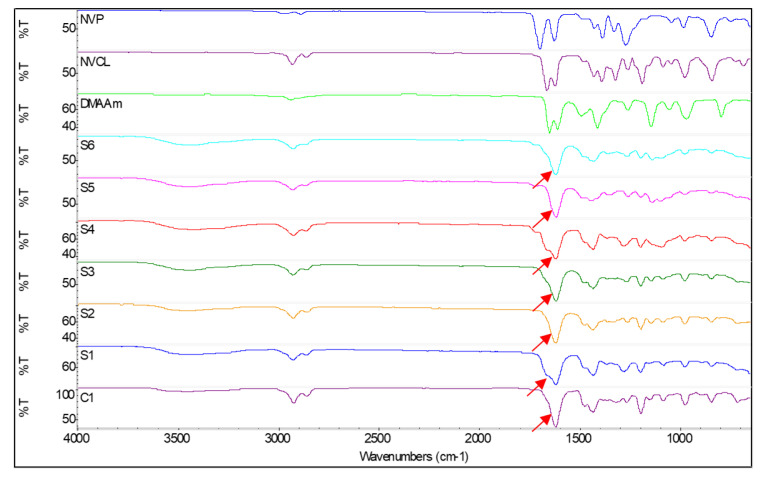
FTIR spectra of the constituent monomer and C1, S1, S2, S3, S4, S5, and S6 samples.

**Figure 4 polymers-14-03155-f004:**
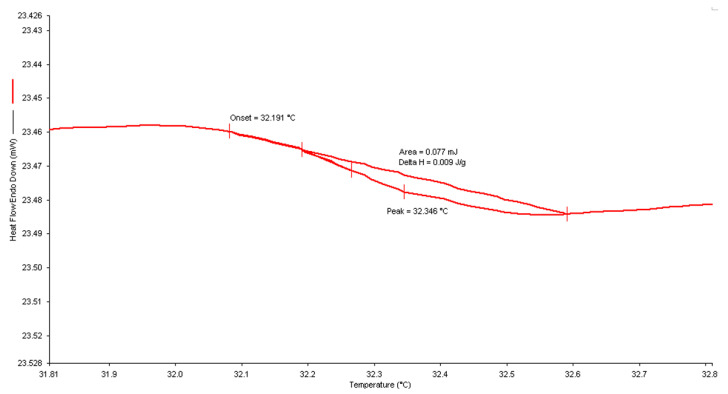
A representative thermogram illustrating the LCST of 10 wt% C1 aqueous solution.

**Figure 5 polymers-14-03155-f005:**
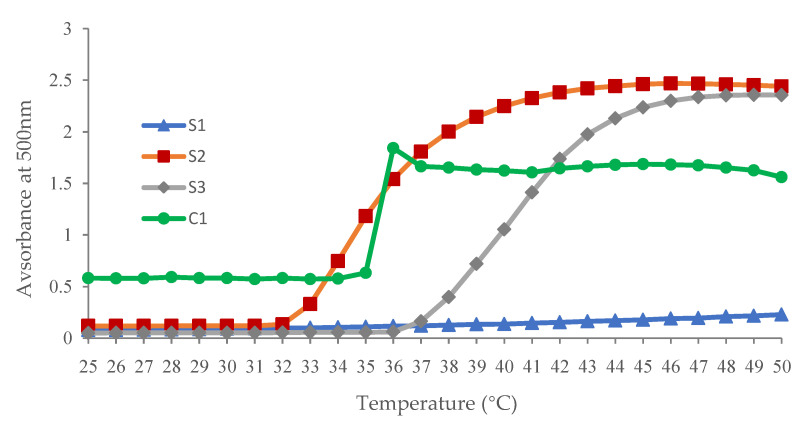
UV spectroscopy illustrating the LCST of C1, S1, S2, and S3 aqueous solutions.

**Figure 6 polymers-14-03155-f006:**
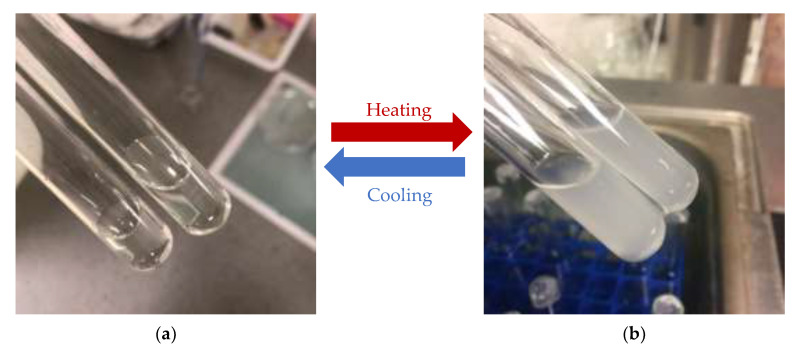
As the temperature rises or falls, the state of the aqueous solution switches between transparent and opaque. (**a**) The state of the aqueous solution of control samples C1 before heating; (**b**) the state of aqueous solution of control samples C1 after heating.

**Figure 7 polymers-14-03155-f007:**
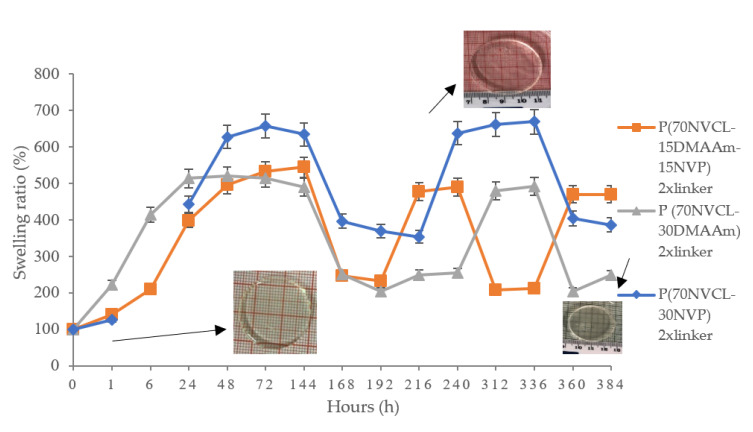
Pulsatile swelling behaviours of chemically crosslinked NVCL/NVP, NVCL/DMAAm, and NVCL/NVP/DMAAm hydrogels with different polymeric compositions and 2 wt% PEGDMA at temperatures between 20 °C and 50 °C.

**Figure 8 polymers-14-03155-f008:**
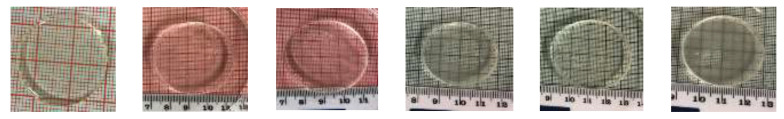
The appearances of the disc samples of S5 at the following times: 0 h, 6 h, 72 h, 216 h, 336 h, and 384 h.

**Table 1 polymers-14-03155-t001:** Name and chemical structure of the selected materials.

Material Names	Chemical Structures
N-vinylcaprolatam (NVCL)	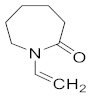
4-(2hydroxyethoxy) phenyl-(2-hydroxy-2-propyl) ketone (Irgacure 2959)	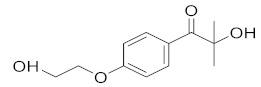
Poly (ethylene glycol) Dimethacrylate(PEGDMA)	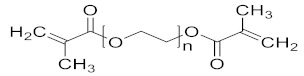
N-vinylpyrrolidone (NVP)	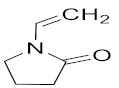
N,N-dimethylacrylammide(DMAAm)	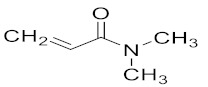

**Table 2 polymers-14-03155-t002:** Compositions of physically and chemically crosslinked xerogels.

	Photoinitiator	Monomer	Comonomers		Crosslinker
Samples	Irgacure 2959 (wt%)	NVCL (wt%)	NVP (wt%)	DMAAm (wt%)	PEGDMA (wt%)
C1	0.1	100	--	--	--
S1	0.1	70	30	--	--
S2	0.1	70	--	30	--
S3	0.1	70	15	15	--
S4	0.1	70	30	--	2
S5	0.1	70	--	30	2
S6	0.1	70	15	15	2

**Table 3 polymers-14-03155-t003:** FTIR bands for NVCL and novel synthesised copolymers.

Functional Group				Shift (cm^−1^)						
	NVCL	DMAAm	NVP	C1	S1	S2	S3	S4	S5	S6
Aliphatic C-H	2929, 2858	2,937	2975, 2886	2923, 2856	2926, 2859	2926, 2858	2926, 2859	2925, 2861	2928, 2864	2926, 2862
C=O	1622	1608	1625	1619	1618	1618	1617	1618	1617	1618
C-N	1480	1490	1488	1473	1471	1474	1473	1471	1483	1473
-CH_2_-	1428	1424	1424	1431	1430	1431	1431	1431	1431	1431
C=C	1662	1648	1696		---	---	---	---	---	---
=CH and =CH_2_	3105, 975	969	981, 843		---	---	---	---	---	---
O-H	---	---	---	3449	3443	3453	3446	3420	3441	3439

**Table 4 polymers-14-03155-t004:** LCST of NVCL-based samples established using DSC, UV-spectroscopy, and cloud point.

Gel Code	DSC (°C)	UV-Spectroscopy (°C)	Cloud Point (°C)
C1	32.2	35	33 ± 0.5
S1	49.4	>50	49.0 ± 0.5
S2	32.3	32	33.0 ± 0.5
S3	36.3	36	37.0 ± 0.5

## Data Availability

Not applicable.

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
