# Peer review of "Lower Critical Solution Temperature Tuning and Swelling Behaviours of NVCL-Based Hydrogels for Potential 4D Printing Applications"

_polymers, 2022, doi:10.3390/polym14153155_

Round 1

Reviewer 1 Report

The article "Lower critical solution temperature tuning and swelling behaviors of NVCL based hydrogels for potential 4D printing applications," presented by Shuo Zhuo, Luke M. Geever, Elaine Halligan, Billy Shu Hieng Tie, and Colette Breheny, investigates the thermal and swelling properties of the prepared NVCL-based polymers. The authors conclude that the polymers developed in this study have variable transition temperatures (LCST), and one of the chemically crosslinked hydrogels exhibits outstanding reversible swelling behaviors, allowing the 4D printing concept to be realized.

In my opinion, the work is presented in an illegible manner. In general, the introductory chapter makes it difficult to determine what the exact purpose of the work was and what novelty this work has compared to those previously published on similar topics.

Notes for authors:

1. Please include a review of the literature for NVCL-based hydrogels obtained thus far, as well as cases where NVP and DMAAm were used as monomers.

2. Why were 0.1 wt% photoinitiator and 2 wt% PEGDMA added? Did the authors carry out optimization reactions for other values of these materials?

3. Figure 2 - please increase the X axis scales. Based on what is visible, the reviewer is unable to determine whether the intense peak at 1620 cm-1 is due to the typical absorption of C = O stretching. Furthermore, why don't the authors explain the presence of a hump at 1620 cm in samples S4 and S6? Please also include the FTIR spectra of the S1, S2, and S3 samples.

4. The 1H NMR spectrum of NVCL and PNVCL produced should be included in the work.

5. Add the LCST values for samples S4, S5, and S6 to Table 3. Please also explain how the LCST value for sample S1 is translated into 4D printing applications.

6. Figure 3 - I believe that including a thermogram illustrating the LCST of the prepared samples S1-S6 is a necessary condition.

7. The authors' conclusion, "the outstanding swellability and tunable phase transition temperature of smart resin have the potential to be used as thermogelling systems in 4D printing techniques, and the printed objects can be used as a candidate to create thermal actuators and sensors," is, in my opinion, far-reaching. In order to achieve a true 4D printing concept, the authors should add the results of applying the obtained preparations to a physical 3D printer to the work.

To summarize, the presented work is not suitable for publication in the Polymers journal in its current form and content.

Reviewer 2 Report

The manuscript is good.

1. Revise the introduction section. Compare the findings of the literature and propose research gaps and objectives of the study.

2. In the materials and method section, provide the names of instruments and model numbers.

3. Section 3 heading "Results" must be "Results and Discussion'

4. Compare the results with literature and provide interpretation.

5. Conclusion section must be written. 

6. The biggest shortfall is no sensitivity analysis. Provide the same.

7. provide the details of the design of the experiment and the appropriate reason to choose the variation of process parameters.

8. Add future research scope and application of research for industry and society.

Reviewer 3 Report

The manuscript by Zhuo et al. deals with synthesis of the termo-responsible (co)polymers based on N-vinyl caprolactam. The authors studied LCST and cloud point temperature values, as well as equilibrium swelling behavior for all synthesized polymers. Also, the suitability of some synthesized materials was established. The manuscript by Zhuo et al. is well-organized, well-written, however I have some comments for the authors:

1.      What is a structure of Irgacure® 2959 photoinitiator? It is hard to understand its influence on physical gel formation due to non-represented structure. If the physical gels are viscosity solutions, so molecular weights of С1, S1-S3 samples must be provided because of great influence of molecular weight on LCST.

2.      The pH of medium also effects on LCST values or swelling behavior. Provide the information of pH values for all conducted experiments.

3.      Why the authors choose 10 % polymer solutions for LCST and cloud point temperature investigation? 1 % or less concentration solutions are usually used for such research.

To my mind, this manuscript may be reconsidered after changes according wit the comments.

Round 2

Reviewer 1 Report

Dear Authors,

Please find my recommendations in the attached document, which is marked in blue. 

Author Response

Please see the attachment, which is marked in green.

Reviewer 3 Report

The authors took in account a significant part of  comments. So, I think that manuscript can be published in the present form.

Author Response

Thank you.

The revised paper was again proofread.